# Evaluation of the Magnetocrystalline Anisotropy of Typical Materials Using MBN Technology

**DOI:** 10.3390/s21103330

**Published:** 2021-05-11

**Authors:** Liting Wang, Cunfu He, Xiucheng Liu

**Affiliations:** Faculty of Materials and Manufacturing, Beijing University of Technology, Beijing 100124, China; wangliting@emails.bjut.edu.cn (L.W.); hecunfu@bjut.edu.cn (C.H.)

**Keywords:** magnetocrystalline anisotropy, magnetic Barkhausen noise, grain orientation, detection parameter

## Abstract

Magnetic Barkhausen noise (MBN) signals in the stage from saturation to remanence of the hysteresis loop are closely correlated with magnetocrystalline anisotropy energy. MBN events in this stage are related to the nucleation and growth of reverse domains, and mainly affected by the crystallographic textures of materials. This paper aims to explore the angle-dependent magnetocrystalline anisotropy energy. Based on the consideration of macroscopic magnetic anisotropy, with the concept of coordinate transformation, a model was firstly established to simulate the magnetocrystalline anisotropy energy (MCE) of a given material. Secondly, the MBN signals in different directions were tested with a constructed experimental system and the characteristic parameters extracted from the corresponding stage were used to evaluate the magnetic anisotropy of the material. Finally, the microstructures of 4 materials were observed with a metallographic microscope. The microtextures of local areas were measured with the electron backscatter diffraction (EBSD) technique. The MBN experimental results obtained under different detection parameters showed significant differences. The optimal MBN detection parameters suitable for magnetic anisotropy research were determined and the experimental results were consistent with the results of MCE model. The study indicated that MBN technology was applicable to evaluate the MCE of pipeline steel and oriented silicon steel, especially pipeline steel.

## 1. Introduction

Ferromagnetic materials such as pipeline steel and silicon steel are widely used in various industries due to their unique mechanical and magnetic properties [1,2]. The magnetic anisotropy of the materials largely affects their overall performances. Ferromagnetic materials are mostly polycrystalline. As one kind of crystalline materials, they have a certain magnetocrystalline anisotropy. The continuous development of key detection technologies such as magnetic Barkhausen noise (MBN) and Electron Backscatter Diffraction (EBSD) has laid a solid foundation for studying the magnetic anisotropy of ferromagnetic materials.

The Barkhausen effect refers to the discontinuous change in the magnetization intensity of a ferromagnetic material in a time-varying magnetic field. The sudden fluctuation of magnetization is caused by the discontinuous movement of the domain wall from one pinning site to the next site as well as the discontinuous rotation of the magnetic domain [3,4]. In other words, MBN signals mainly come from irreversible domain wall motion and irreversible domain rotation. The intensity of MBN jumps generated by the former is large enough to overwhelm that generated by the latter [5]. Therefore, in previous studies on the magnetic anisotropy of MBN signals, the magnetization mechanism was ascribed to the irreversible motion of 180° domain wall. In this case, the direction of the magnetic easy axis was usually along the direction where the characteristic value of MBN signals was the highest [6,7]. In recent years, some researchers [8,9,10] have focused on smaller MBN jumps caused by magnetic domain rotation. Espina-Hernández et al. [8] experimentally proved that the angular dependency of magnetocrystalline anisotropy energy (MCE) was closely correlated with multi-angle MBN signals in the saturation-to-remanence stage of the hysteresis loop in pipeline steels. Therefore, the magnetic anisotropy in the MBN signals with irreversible domain rotation has become a new research direction.

In order to describe MBN signals, a large number of mathematical models have been developed. Existing MBN models, including the ABBM model [11,12,13], J-A hysteresis model [14,15] and multi-angle MBN energy model [16,17,18], were mainly focused on strong MBN jumps in the area around the coercive force point where domain wall motion dominated. However, existing theoretical models for the low-intensity MBN jumps related to the nucleation and growth of reverse domain in the saturation-to-remanence stage of the hysteresis loop cannot provide satisfactory simulation results. So far, only a Mexican team [9,19] has developed a model for MBN signals generated by the nucleation and growth of the reverse domain in this stage and interpreted the observed strong correlation between MBN signals and MCE.

The intrinsic microstructures (grain boundaries, grain orientations and grain sizes) of various ferromagnetic materials are different, thus leading to the completely different magnetic properties of the materials. During the preparation of the material, a variety of methods can be used to adjust the microstructure and improve the magnetic properties of the material, such as crystallographic texture enhancement [20], controlling cooling speed [21], and adjustment with a strip steel casting process [22]. MBN is sensitive to microstructure parameters, including grain size [23], grain boundary [24], grain orientation [25] and carbon content [26], and also has high sensitivity to stress and residual stress [27]. The measurement and analysis of the MBN envelope provides relevant information about the magnetic properties of different materials and is affected by different magnetization mechanisms. However, in actual MBN tests, the MBN envelope is closely related to the measurement method and detection parameters [28]. Therefore, according to various experimental schemes based on MBN technology, the evaluation results of the magnetic anisotropy of the same material are often different.

At present, the MBN signals affected by MCE have been seldom reported. Only a generalized model has been proposed to describe the randomness of MBN signals in the saturation-to-remanence stage. The magnetocrystalline anisotropy energy model or the influence of experimental parameters on model verification results has not been reported. Based on the generalized model, a theoretical model of angle-dependent MCE was firstly proposed in this study. Then, the microstructure parameters of the materials were measured with EBSD technology and introduced into the theoretical model. Through multi-angle MBN detection experiments under different detection parameters, the applicability and accuracy of the model were verified and the optimal MBN detection parameters suitable for magnetic anisotropy research were finally determined. The developed model and corresponding technical scheme can be used in the MBN evaluation of the magnetic anisotropy of ferromagnetic materials.

## 2. Theoretical Analysis

A model for the nucleation and growth of reverse domains was proposed [29]. The most possible origin of reverse domain nucleation are grain boundaries or lamellar precipitates. When the strength of the magnetic field is low, the magnetization vectors of adjacent grains cannot rotate from their easy axis to the direction completely aligned with the magnetic field. The magnetization vector component perpendicular to the grain boundary (GB) is generally discontinuous, so that magnetic free poles appear at the GBs. The nucleation of reverse domains is related to magnetic free poles at the GBs. The nucleation model of a reverse domain at a grain boundary is shown in Figure 1. The magnetic free pole density *ω** at the *i*-th grain boundary can be expressed as Equation (1):(1)$$\omega_{i}{}^{*}=J_{s}\left(\cos\theta_{i1}-\cos\theta_{i2}\right)=M_{s}\mu_{0}\left(\cos\theta_{i1}-\cos\theta_{i2}\right)$$
where *M_s_* is the saturation magnetization of the single crystal, *θ*_1_ and *θ*_2_ are the angles formed by the magnetization vectors of adjacent grains and the normal to GB; and *μ*_0_ is the vacuum permeability.

When the magnetic state of the material is changed from the saturated state to the remanence state, the magnetization direction of the grains on both sides of the GB rotates from the direction of the magnetic field to the direction of the easy axis. Due to the distribution of misorientation between the grains, magnetic free poles appear at the GBs. Then, small reversed domains are nucleated at GBs in order to reduce the extra magnetostatic energy generated by these magnetic free poles. The values of *ω***_i_* under the alternating magnetic field *H* of different angles are different, so the average magnetic free pole density in polycrystalline can be expressed as Equation (2) [19]:(2)$$\bar{\omega }\left(\eta |H_{\eta }\right)=\frac{1}{N_{GB}}\sum_{i=1}^{N_{GB}}\left|\omega^{*}{}_{i}\left(\eta |H_{\eta },g_{GB},\theta_{i1},\theta_{i2}\right)\right|$$
where *H_η_* is the magnetic field value at each angular position *η*; *g_GB_* is the orientation of the GB; *N_GB_* is the number of GBs.

The MBN jumps are produced by the nucleation and growth of the reverse domains when the material goes from saturation to remanence. Based on the consideration of the growth speed of the reverse domain from a nucleation point at GBs, grain size, and carbon content, the MBN energy in this stage can be expressed as Equation (3) [9]:(3)$$E_{MBN}\left(\eta |H_{\eta },p,d_{g}\right)\propto \frac{2\rho J_{s}\bar{\chi_{d}}}{{\bar{d_{g}}}^{2}\mu_{o}}\left(H-H_{n}-H_{g}\right)\bar{\omega }\left(\eta |H_{\eta },p\right)$$
where *ρ* is the resistivity; $\bar{\chi_{d}}$ is average differential magnetic susceptibility; $\bar{d_{g}}$ is average grain size; *p* is the pearlite content; *H* is the applied field; and *H_n_* and *H_g_* are, respectively, the critical magnetic field strength for generating the reverse domain and the threshold magnetic field required for moving the domain wall after nucleation.

MCE is closely related to the orientation of the magnetization vector relative to the crystal axis. MCE has a direct relationship with the crystallographic texture of the material. The density of magnetic free poles is determined by the grain-to-grain misorientation defined by the crystallographic texture, so the density of magnetic free poles is proportional to MCE [30]. Based on Equation (3) where *E_MBN_* is proportional to $\bar{\omega }$, the following relationship can be obtained:(4)$$E_{MBN}\propto \bar{\omega }\propto MCE$$

Equation (4) shows that a higher number of average magnetic free poles in the polycrystalline corresponds to a larger number of reverse domains and stronger MBN signals. In other words, the MBN events generated in this stage are mainly affected by the grain orientation distribution of the material.

## 3. Magnetocrystalline Anisotropy Energy Model

In this study, based on the concept of coordinate transformation, the material macroscopic reference coordinate system is combined with the microscopic grain orientation and a model is then established to simulate the MCE of a given grain orientation.

As shown in Figure 2, the rolling direction of the material is set as the reference direction and the angle between the applied magnetic field *H* and the reference direction is *η*. The magnetization direction φ in the material varies with the applied magnetic field strength.

The total magnetic energy in the system is expressed as Equation (5):(5)$$E\left(\eta ,\varphi \right)=K_{1}\sin^{2}\varphi \cos^{2}\varphi -\mu_{0}M_{s}H\cos\left(\varphi -\eta \right)$$
where *K*_1_ is the magnetocrystalline anisotropy constant.

According to the principle of thermodynamic balance, φ is determined by minimizing the total magnetic energy on both sides of the GB. For each angular position *η* and a given value of *H*, a corresponding value of φ can be obtained and its steady state condition is expressed as:(6)$$\left\{ \begin{array}{l} \frac{dE\left(\eta ,\varphi \right)}{d\varphi }=0\\ \frac{d^{2}E\left(\eta ,\varphi \right)}{d{\left(\varphi \right)}^{2}}>0 \end{array}\right.$$

According to the magnetic properties of iron single crystal, parameters are set as *H* = 5 × 10^4^ (A/m), *K*_1_ = 4.8 × 10^4^ (J/m^3^), and *M_s_* = 1.71 × 10^6^ (A/m). Through changing the angle *η* of the applied magnetic field with the step size of 5°, Equation (6) can be solved to obtain the magnetization direction φ (star point) corresponding to each angular position *η*, as shown in Figure 3. The point intersecting the straight line in the figure represents the magnetization direction, which is the same as the magnetic field direction. However, at the remaining angular positions, the *δ* value represents the angle between the magnetization direction and the magnetic field direction. The smaller the *δ* value is, the closer the magnetization direction is to the magnetic field direction.

As shown in Figure 4, there is a macroscopic reference coordinate system *O*-*XYZ* and a cubic crystal coordinate system *O*′-*x*′ *y*′ *z*′ in the space. Based on the concept of coordinate transformation, the crystal coordinate system in the initial orientation is rotated in the order of $\varphi_{1}$ (0 ≤ $\varphi_{1}$≤ 2π), Φ (0 ≤ Φ ≤ π) and $\varphi_{2}$ (0 ≤ $\varphi_{2}$ ≤ 2π) to obtain any grain orientation in space ($\varphi_{1}$, Φ, $\varphi_{2}$). These three independent angles are called Euler angles. The grain orientation *g* after being rotated by the Euler angle is expressed as Equation (7) [31]:(7)$$\begin{array}{l} g=\left[\begin{array}{ccc} \begin{array}{l} \cos\varphi_{2}\\ -\sin\varphi_{2}\\ 0 \end{array} & \begin{array}{l} \sin\varphi_{2}\\ \cos\varphi_{2}\\ 0 \end{array} & \begin{array}{l} 0\\ 0\\ 1 \end{array} \end{array}\right]\left[\begin{array}{ccc} \begin{array}{l} 1\\ 0\\ 0 \end{array} & \begin{array}{l} 0\\ \cos\phi \\ -\sin\phi \end{array} & \begin{array}{l} 0\\ \sin\phi \\ \cos\phi \end{array} \end{array}\right]\left[\begin{array}{ccc} \begin{array}{l} \cos\varphi_{1}\\ -\sin\varphi_{1}\\ 0 \end{array} & \begin{array}{l} \sin\varphi_{1}\\ \cos\varphi_{1}\\ 0 \end{array} & \begin{array}{l} 0\\ 0\\ 1 \end{array} \end{array}\right]\\ \;\;=\left[\begin{array}{ccc} \begin{array}{l} \cos\varphi_{1}\cos\varphi_{2}-\sin\varphi_{1}\cos\phi \sin\varphi_{2}\\ -\cos\varphi_{1}\sin\varphi_{2}-\sin\varphi_{1}\cos\phi \cos\varphi_{2}\\ \sin\varphi_{1}\sin\phi \end{array} & \begin{array}{l} \sin\varphi_{1}\cos\varphi_{2}+\cos\varphi_{1}\cos\phi \sin\varphi_{2}\\ -\sin\varphi_{1}\sin\varphi_{2}+\cos\varphi_{1}\cos\phi \cos\varphi_{2}\\ -\cos\varphi_{1}\sin\phi \end{array} & \begin{array}{l} \sin\phi \sin\varphi_{2}\\ \cos\varphi_{2}\sin\phi \\ \cos\phi \end{array} \end{array}\right] \end{array}$$

When many grains in polycrystalline are oriented in one or some orientation positions, this situation is called the texture. For example, grains in oriented silicon steel with Goss texture are mostly arranged in the vicinity of the rolling direction. In the production and processing of ferromagnetic materials, after cold and hot rolling, the crystal structure of the materials show the texture phenomenon to different degrees, which cause anisotropy in the structure and performances of materials. In this study, individual grain orientation in the selected area was analyzed with EBSD technology.

Taking a given grain orientation *g*_1_ = ($\varphi_{1}$, Φ, $\varphi_{2}$) = (0°, 0°, 45°) as an example, the components of the magnetization vector in the reference coordinate system and the crystal coordinate system are respectively *x*, *y*, *z* and *x*′, *y*′, *z*′ and the matrix *g*_1_ is expressed as Equation (8):(8)$$g_{1}=\left[\begin{array}{ccc} \begin{array}{l} \cos\varphi_{2}\\ -\sin\varphi_{2}\\ 0 \end{array} & \begin{array}{l} \sin\varphi_{2}\\ \cos\varphi_{2}\\ 0 \end{array} & \begin{array}{l} 0\\ 0\\ 1 \end{array} \end{array}\right]$$

Then the components of the magnetization vector in the crystal coordinate system are expressed as Equation (9):(9)$$\left\{ \begin{array}{l} x^{’}=\cos\varphi_{2}x+\sin\varphi_{2}y\\ y^{’}=-\sin\varphi_{2}x+\cos\varphi_{2}y\\ z^{’}=z \end{array}\right.$$

Based on the macroscopic magnetization direction obtained by Equation (6), the components of the magnetization vector in the reference coordinate system at any given angular position *η* can be expressed as Equation (10):(10)$$\left\{ \begin{array}{l} x=\left|\vec{M_{s}}\right|\cos\varphi \\ y=\left|\vec{M_{s}}\right|\sin\varphi \\ z=0 \end{array}\right.$$

By substituting Equation (10) into Equation (9), the components of the magnetization vector in the crystal coordinate system can be determined as Equation (11):(11)$$\left\{ \begin{array}{l} x^{’}=\left|\vec{M_{s}}\right|\cos\varphi_{2}\cos\varphi +\left|\vec{M_{s}}\right|\sin\varphi_{2}\sin\varphi =\left|\vec{M_{s}}\right|\cos\left(\varphi_{2}-\varphi \right)\\ y^{’}=-\left|\vec{M_{s}}\right|\sin\varphi_{2}\cos\varphi +\left|\vec{M_{s}}\right|\cos\varphi_{2}\sin\varphi =-\left|\vec{M_{s}}\right|\sin\left(\varphi_{2}-\varphi \right)\\ z^{’}=0 \end{array}\right.$$

The magnetocrystalline anisotropy energy of a single crystal is expressed as Equation (12) [30]:(12)$$F_{k}=K_{0}+K_{1}\left(\alpha_{1}^{2}\alpha_{2}^{2}+\alpha_{2}^{2}\alpha_{3}^{2}+\alpha_{1}^{2}\alpha_{3}^{2}\right)+K_{2}\alpha_{1}^{2}\alpha_{2}^{2}\alpha_{3}^{2}$$
where *K*_1_ and *K*_2_ are magnetocrystalline anisotropy constants; *K*_0_ is an angle-independent constant representing the isotropic component; *α*_1_, *α*_2_ and *α*_3_ are the direction cosines of *M_s_* with respect to the three crystal axes. According to Equation (11), the following relationship can be obtained as Equation (13):(13)$$\left\{ \begin{array}{l} \alpha_{1}=\cos\left(\varphi_{2}-\varphi \right)\\ \alpha_{2}=-\sin\left(\varphi_{2}-\varphi \right)\\ \alpha_{3}=0 \end{array}\right.$$

Finally, the MCE of cubic crystal with Euler angle *g*_1_ can be expressed as Equation (14):(14)$$F_{k1}=K_{0}+K_{1}\left[\cos^{2}\left(\varphi_{2}-\varphi \right)\sin^{2}\left(\varphi -\varphi_{2}\right)\right]$$

With Matlab software, the MCE at any angular position *η* can be obtained, as shown in Figure 5.

In order to verify the applicability of the model, taking another grain orientation *g*_2_ = ($\varphi_{1}$, Φ, $\varphi_{2}$) = (0°, 45°, 0°) as an example, the MCE can be expressed as Equation (15), as shown in Figure 6a.
(15)$$F_{k2}=K_{0}+K_{1}\left[\cos^{2}\varphi \sin^{2}\varphi +\cos^{2}\phi \sin^{2}\phi \sin^{4}\varphi \right]$$

Even in strong texture materials such as oriented silicon steel, there is more than one grain orientation. It is assumed that 60% and 40% of grains are, respectively, in the orientation *g*_1_ and orientation *g*_2_ inside the material. By solving the weighted average of the MCE of each grain, the MCE of the polycrystal can be obtained (Figure 6b).

When *H* is set as 10 × 10^4^ (A/m), with the increase in the strength of the applied magnetic field, the magnetization direction φ corresponding to each angular position can be obtained (Figure 7).

The comparison results of Figure 3 and Figure 7 are summarized below. At the angular positions of 0°, 45°, 90°, 135°, 180°, 225°, 270°, 315° and 360°, the magnetization direction rotates to the same direction as the magnetic field. However, at the other angular positions, increasing the applied magnetic field strength reduces the *δ* value, indicating that the driving force of the external magnetic field is stronger at this time and makes the magnetization direction closer to the direction of the magnetic field.

Similarly, taking the grain orientations *g*_1_, *g*_2_, and 60% *g*_1_ + 40% *g*_2_ as examples, the simulation results of MCE are shown in Figure 8. It can be seen from Figure 8 that increasing the magnetic field strength does not change the positions of the magnetic hard axis and magnetic easy axis determined by the crystal structure, namely, the positions of the short and long axes in the MCE pole diagram. At 9 special angular positions, since the magnetization direction has not changed, the MCE value has not changed. At other angular positions, the magnetization direction is closer to the magnetic field direction, thus changing the MCE value.

## 4. Experimental Measurement Method

### 4.1. MBN Test

The experimental setup shown in Figure 9 was used to test MBN signals. The entire experimental system was controlled by the LabVIEW program installed on the host computer. Sine wave signals were generated with an excitation board and output to the KEPCO BOP100-4ML bipolar power amplifier. After the amplification, the signals entered the excitation coil wound on the top of the U-shaped electromagnet. A detection coil was arranged on the central axis of the U-shaped magnetic circuit to receive the MBN signals. The detection coil was made of 2000 turns of varnished wire with a wire diameter of 0.05 mm, an outside diameter of 5.4 mm, an inner diameter of 2 mm, and a height of 10 mm and filled with ferrite cores of the same height. The output voltage signals of the detection coil were collected by the NI-PXIe-6376 multi-channel acquisition card.

Three types of electromagnets were designed, and all the electromagnets had a U-shaped ferrite core with an excitation coil wound on the top. The inner span C of the three U-shaped ferrite cores was different. The size parameters are shown in Table 1.

In order to measure MBN signals in different directions, the sensor base shown in Figure 9b was produced by 3D printing and had card slots distributed at equal angular intervals of 10°. By manually rotating the MBN detection sensor and placing it in the corresponding card slot, the MBN signal detection in different directions was implemented.

Four steel plates of different materials were tested. The length and width of 30SQG120 oriented silicon steel, B50A470 non-oriented silicon steel, and X60 and X70 pipeline steels are the same (200 mm × 200 mm) and their heights are 0.3 mm, 0.5 mm, 2 mm, and 2 mm, respectively. For a given material, the results of the magnetic anisotropy tested based on MBN signals were affected by a variety of testing parameters. In this study, the magnetic circuit span, excitation frequency and excitation field amplitude were selected as the three main influencing factors. A total of 12 sets of testing parameters were designed (Table 2).

The first group of detection parameters were taken as an example to illustrate the signal processing and feature parameter extraction methods. The output voltage signals of the MBN detection coil were band-pass filtered (10~50 kHz) with the 4th-order Butterworth digital filter and the MBN signals were smoothed to obtain the MBN envelope curve with the moving average method. After the background noise threshold point of MBN signals was set to be 0.1 mV, the starting point A of the MBN envelope curve was determined. Then the first intersection point of the 75% envelope peak and the MBN envelope curve was selected as the cutoff point B. Then root mean square RMS in the section from point A to point B was extracted as the characteristic parameter.

In the test, the rolling direction of steel plates was set as the reference direction and the angle of the excitation field *H* relative to the reference direction was gradually changed with a step of 10° (Figure 9c). A total of 5 repeated MBN detection experiments were performed at each angle *θ* and the characteristic parameters obtained in all the experiments were averaged to analyze the magnetic anisotropy of the materials. With the third-order Fourier series expansion, the characteristic parameter RMS is fitted as Equation (16):(16)$$RMS=a_{0}+\sum_{n=1}^{3}\left[a_{n}\cos\left(n\omega \eta \right)+b_{n}\sin\left(n\omega \eta \right)\right]$$
where *a*, *b*, and *ω* are undetermined coefficients and *η* is the angle of the excitation field *H* relative to the reference direction. Finally, the results of the anisotropy are obtained in the form of the MBN characteristic parameter pole diagram, as shown in Figure 10.

### 4.2. Microstructure

After the MBN test was completed, the 4 materials were cut into the samples with a size of 30 × 30 mm. Metallographic samples were prepared according to standard procedures. The microstructures are shown in Figure 11.

The microstructures of B50A470 non-oriented silicon steel, X60 and X70 pipeline steel showed a random distribution pattern (Figure 11), indicating that the microstructures of these three materials did not affect their anisotropy. However, in 30SQG120 oriented silicon steel, the grains were elongated obviously along the rolling direction to form a typical fibrous structure, indicating that the anisotropy of 30SQG120 oriented silicon steel was strong.

### 4.3. EBSD Testing

In global texture measurements, all the grains in a polycrystalline are analyzed as a whole, whereas microtexture measurements can provide specific information such as grain orientation, grain size and misorientation between adjacent grains in the polycrystalline. The rapid developed EBSD technology has become the main means of micro-texture detection. In this study, JSM-7900F thermal field emission scanning electron microscope equipped with EBSD analysis software was used to perform the orientation imaging analysis on the local area of the materials. Taking the point *O* shown in Figure 9c as the center point, a sample with the size of 10 × 10 mm was cut from 4 materials by a wire cutting method for microtexture measurements. In order to accurately estimate the MCE of the tested sample with the obtained texture data, it is necessary to adjust the scan step length to determine the appropriate number of grains in the observation area. Table 3 provides the number of grains determined by EBSD microtexture measurements.

Figure 12 shows the orientation imaging of four materials. The grain orientation of 30SQG120 oriented silicon steel had a preference phenomenon. The lattice orientation of each grain was mostly the same, indicating that there was a strong texture inside the material. The grain orientation of B50A470 non-oriented silicon steel seemed to be randomly distributed and its texture phenomenon was not obvious. The orientation diagrams of X60 and X70 pipeline steels were respectively green and blue, displaying the texture phenomenon to a certain degree.

According to the method described above, with the orientation of each grain represented by different colors in Figure 12, the MCE model was obtained. By averaging the contribution of individual grains with a direct method, the MCE results based on the consideration of the direction of the external magnetic field were finally obtained (Figure 13).

## 5. Analysis and Discussion

The MBN envelope curve is closely related to the detection parameters, so the magnetic anisotropy results obtained under different detection parameters are not completely the same. In order to ensure that the MBN test results are consistent with the model results, it is necessary to determine the optimal detection parameters for exploring the magnetic anisotropy.

### 5.1. Influences of Detection Parameters on Experimental Results

In order to eliminate the influences of the parameter dimension on the image comparison, the results of the MCE model and the MBN results obtained from the experiment were normalized. Figure 14 shows a qualitative comparison between the experimental results obtained under different detection parameters and the results of the MCE model. The experimental results obtained under different detection parameters showed significant differences and the obtained main characteristics of magnetic anisotropy also showed angular deviations. In X60 pipeline steel, X70 pipeline steel and 30SQG120 oriented silicon steel, the experimental results corresponding to the 10th, 4th and 3rd groups of test parameters were largely consistent with the results of the MCE model. In order to quantitatively describe the correlation between the two methods, the angles *θ*_y_ and *θ*_n_ between the reference direction and the long and short axes in the pole diagram were extracted to represent the direction cosine of the magnetic easy and hard axes. Table 4 summarizes the characteristic parameters *θ*_y_ and *θ*_n_ related to magnetic anisotropy extracted from Figure 14.

According to the analysis results in Table 4, the maximum deviation of the direction cosine of the magnetic hard axis obtained by the two methods is 14°. Since the Fourier series method was used to smooth the results of MBN experiments, this small discrepancy was acceptable.

In B50A470 non-oriented silicon steel, it was difficult to find a group of detection parameters which were consistent with the simulated values. Non-oriented silicon steel has a nearly isotropic crystal texture, and there is residual stress inside the material. The measured MBN anisotropy is the consequence of stress-induced magnetic anisotropy and magnetocrystalline anisotropy. The MCE model can only reflect the magnetocrystalline anisotropy of the material. Therefore, it is difficult to evaluate the MCE of non-oriented silicon steel with MBN technology. Oriented silicon steel is a strong texture material and the direction of the magnetic easy axis produced by the two mechanisms in oriented silicon steel are similar. Therefore, compared with non-oriented silicon steel, MBN technology can still be used to evaluate MCE.

### 5.2. Evaluation of MCE with MBN Technology

Under different testing parameters, the valid magnetic field strength and testing depth in a material are different. Therefore, it is necessary to compare the absolute values of MCE of the four materials obtained with the two methods under a group of fixed detection parameters (Figure 15). The optimal detection parameters of each material are different. The set of parameters selected in Figure 15 were suitable for evaluating the pipeline steel material and therefore the differences in the two results of the 30SQG120 oriented silicon steel could be understood. As for non-oriented silicon steel (B50A470), although the pattern obtained by the two methods exhibited four-petal butterfly type, the resulting hard axes (short axis) were approximately perpendicular to each other. If the model results were rotated 90 degrees clockwise, the results obtained by the two methods had strong correlation. Excepting B50A470 non-oriented silicon steel, the other three materials showed the consistent relative values of MCE when the two methods were respectively used. Based on the aforementioned qualitative comparison results, it was confirmed that MBN signals extracted in the section from point A to point B could be used to evaluate the MCE of oriented silicon steel and pipeline steel. The MBN signals in this stage were also closely correlated with the crystallographic structure. This conclusion was consistent with the previous results [32].

The magnetic anisotropy of a material is mainly affected by three mechanisms: average magnetocrystalline anisotropy, processing (texture, dislocation packing, etc.) and stress-induced magnetic anisotropy. Inside the silicon steel material produced by cold rolling, crystals are severely deformed and elongated in the rolling direction, thus making the texture direction close to the rolling direction. In addition, residual stress is generated during the deformation of grains and the residual stress in the rolling direction is significantly larger than that perpendicular to the rolling direction. The experimental results of silicon steel materials included the effect of residual stress on magnetic anisotropy. Therefore, the MCE evaluation method based on MBN technology was more suitable for pipeline steel materials than silicon steel.

## 6. Conclusions

In this study, a theoretical model of angle-dependent MCE was firstly proposed and then the applicability and accuracy of the model were verified through EBSD technology and MBN test experiments. In addition, the optimal MBN detection parameters suitable for magnetic anisotropy research were obtained. The main conclusions are drawn as follows:(1)Based on the concept of coordinate transformation, the material macroscopic reference coordinate system was combined with the microscopic grain orientation. The EBSD technology was used to measure the micro-texture of the local area, and a model was established to simulate the magnetocrystalline anisotropy of given materials.(2)With MBN technology, the magnetic anisotropy of materials was evaluated. The obtained experimental results were in good agreement with the results of the MCE model, indicating that the MBN technology could be used to evaluate the MCE of pipeline steel and oriented silicon steel.(3)The MBN experimental results obtained under different detection parameters were significantly different, so it is necessary to determine the optimal detection parameters for exploring magnetic anisotropy.(4)Non-oriented silicon steel has a nearly isotropic crystallographic texture and it is difficult to predict its MCE with MBN technology. Due to the residual stress in silicon steel materials, the MCE evaluation method based on MBN technology was more suitable for pipeline steel than silicon steel.

## Figures and Tables

**Figure 1 sensors-21-03330-f001:**
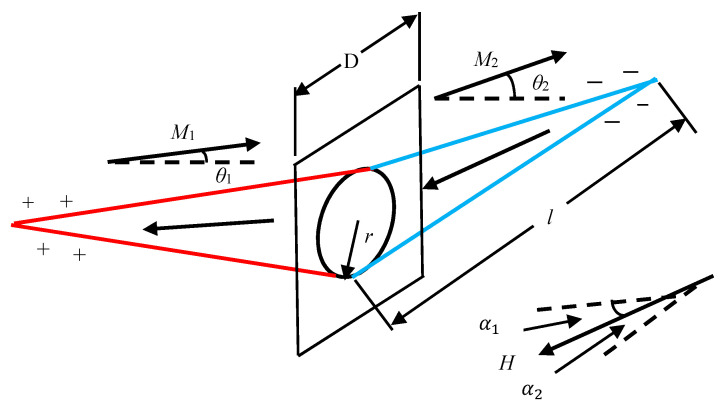
Nucleation model of a reverse domain at a grain boundary.

**Figure 2 sensors-21-03330-f002:**
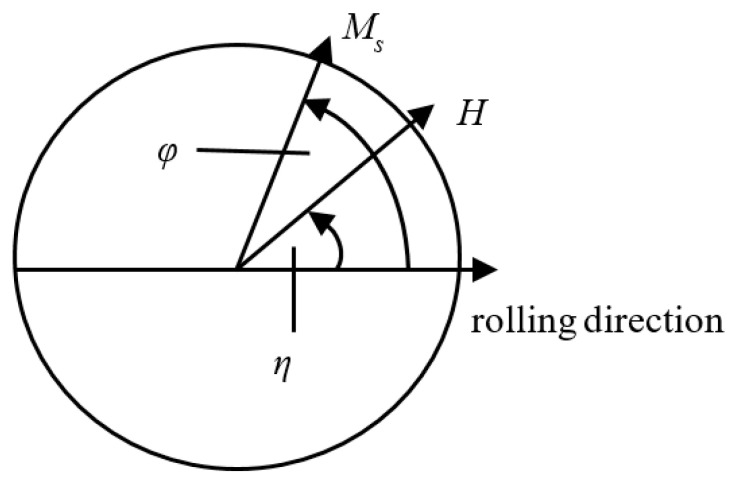
Magnetization direction under a given applied magnetic field strength.

**Figure 3 sensors-21-03330-f003:**
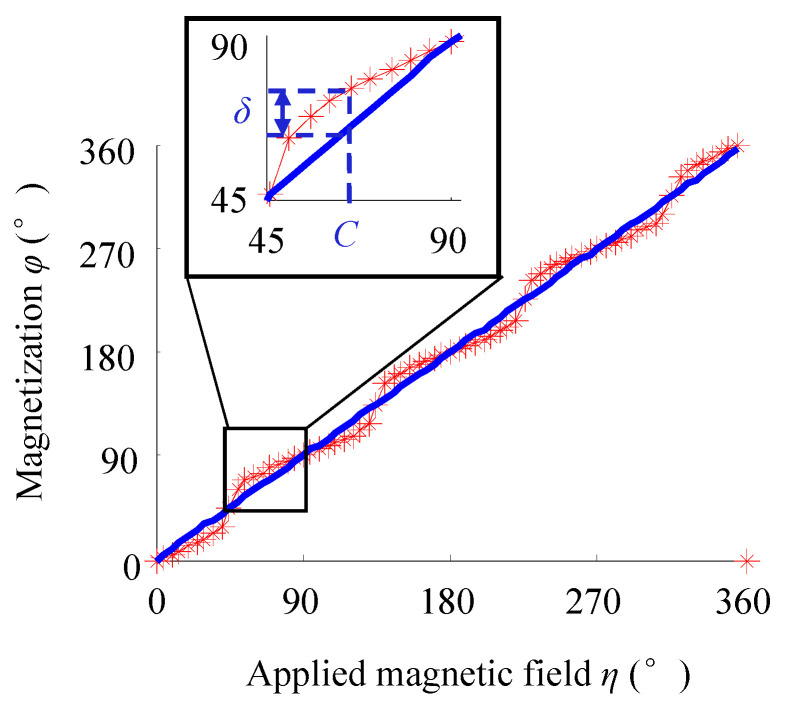
Magnetization direction under the applied magnetic field with different angular positions *η*.

**Figure 4 sensors-21-03330-f004:**
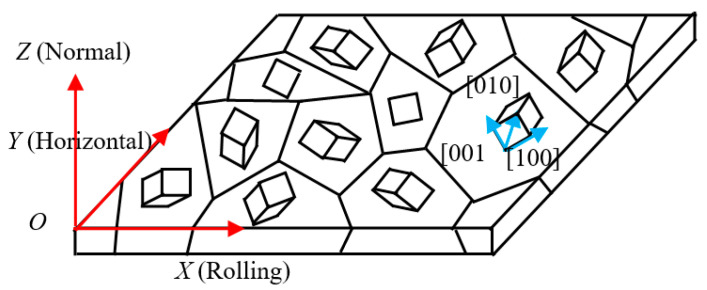
Schematic diagram of general grain orientation inside polycrystalline materials.

**Figure 5 sensors-21-03330-f005:**
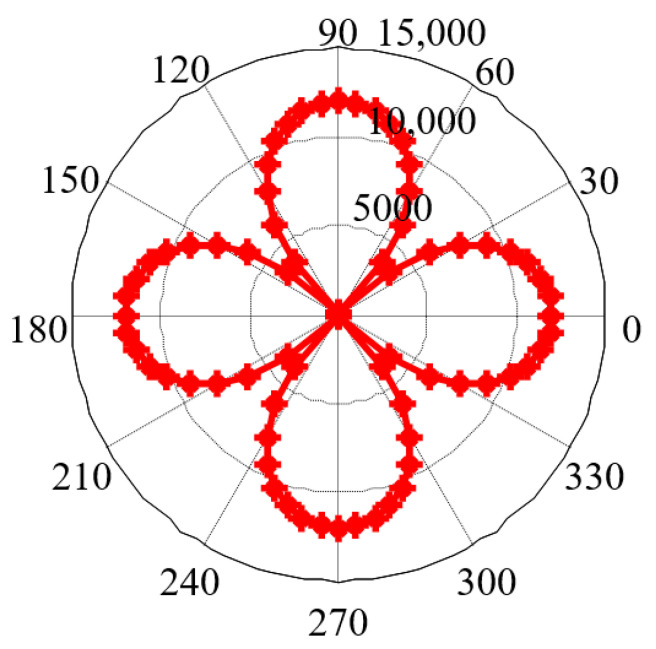
Simulation results of MCE (*g*_1_).

**Figure 6 sensors-21-03330-f006:**
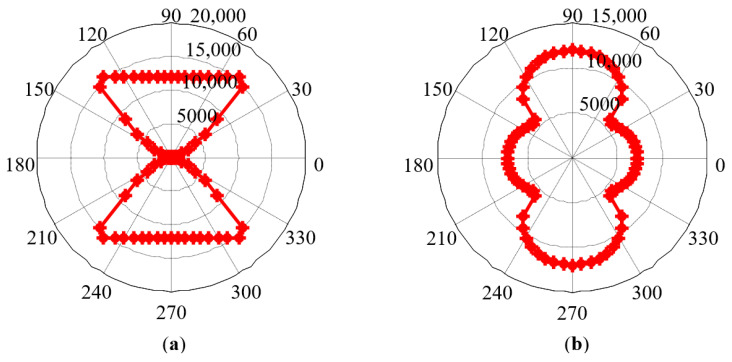
Simulation results of MCE: (**a**) grain orientation *g*_2_; (**b**) grain orientation 60%*g*_1_ + 40%*g*_2_.

**Figure 7 sensors-21-03330-f007:**
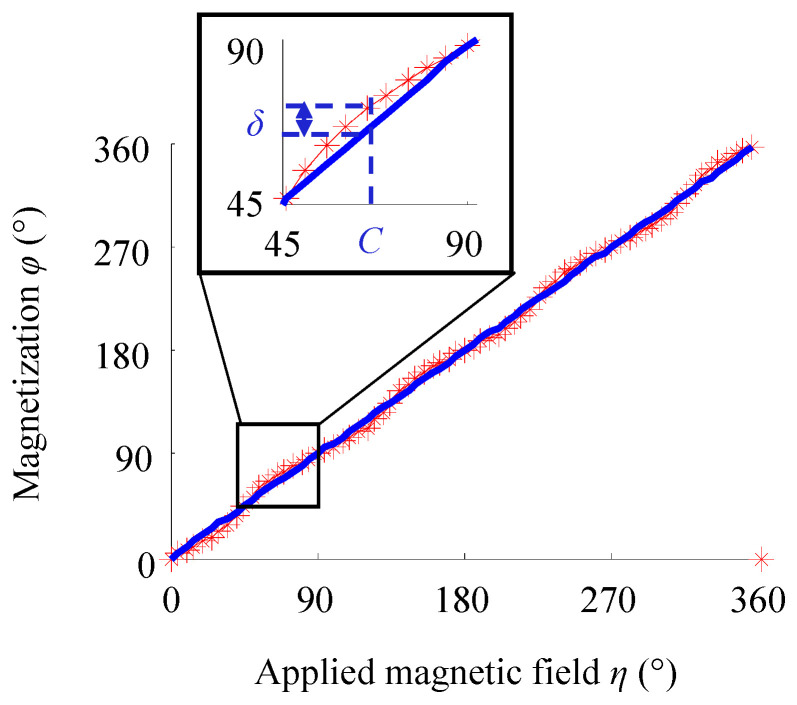
Magnetization direction at each angular position with the increase in the applied magnetic field strength.

**Figure 8 sensors-21-03330-f008:**
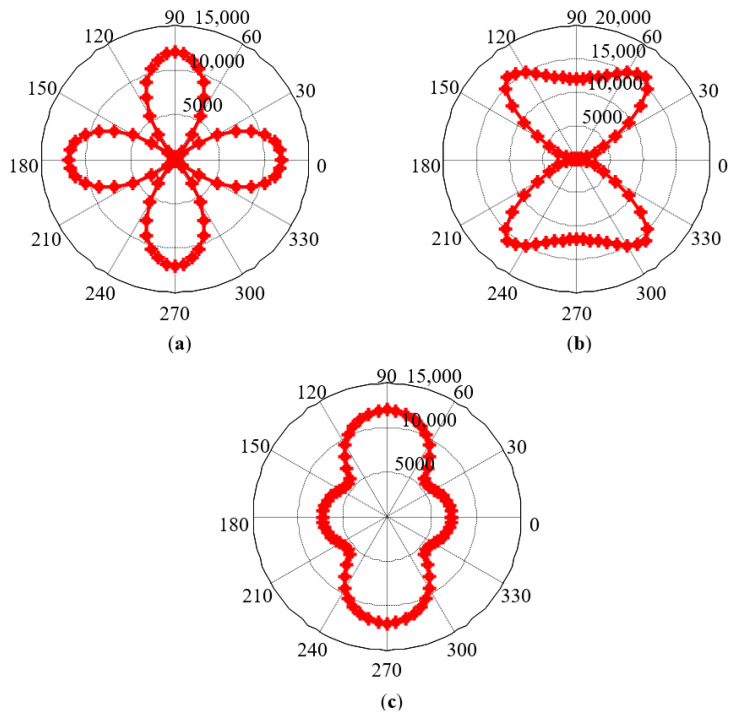
Simulation results of MCE under the condition of the increasing magnetic field strength: (**a**) grain orientation *g*_1_; (**b**) grain orientation *g*_2_; (**c**) grain orientation 60% *g*_1_ + 40% *g*_2_.

**Figure 9 sensors-21-03330-f009:**
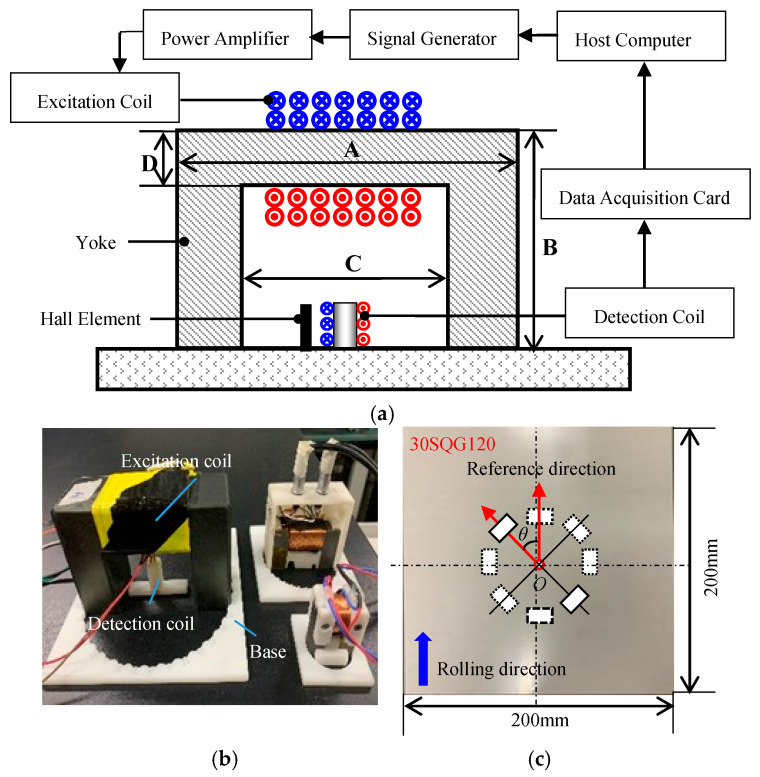
Experimental setup: (**a**) flow chart of the experimental system, (**b**) sensors, and (**c**) detection diagram.

**Figure 10 sensors-21-03330-f010:**
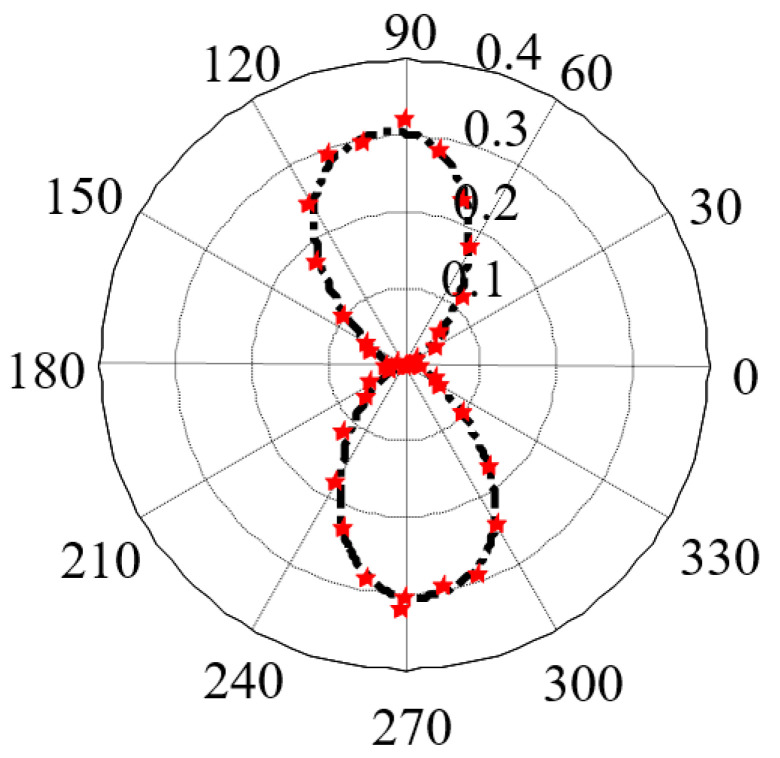
Magnetic anisotropy measured in X60 pipeline steel.

**Figure 11 sensors-21-03330-f011:**
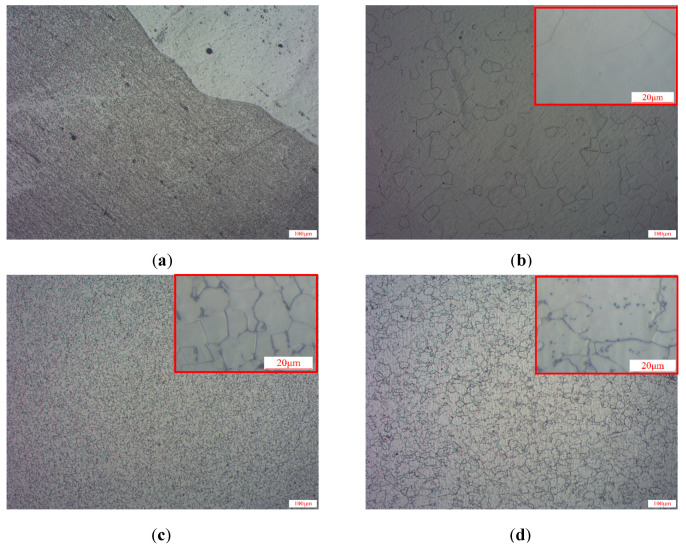
Optical micrographs of the microstructures of 4 materials: (**a**) 30SQG120, (**b**) B50A470, (**c**) X60, and (**d**) X70.

**Figure 12 sensors-21-03330-f012:**
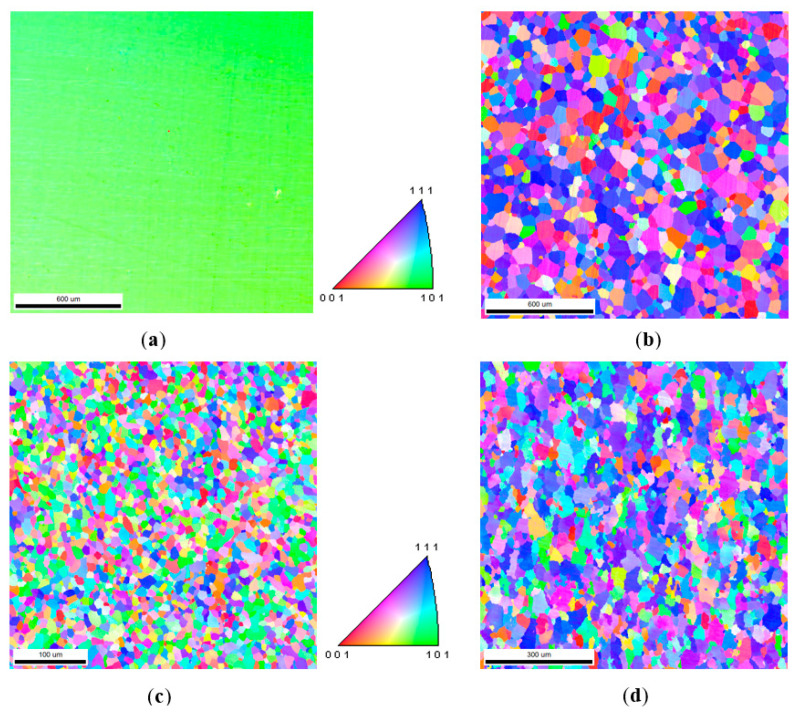
Orientation maps of 4 materials: (**a**) 30SQG120, (**b**) B50A470, (**c**) X60, and (**d**) X70.

**Figure 13 sensors-21-03330-f013:**
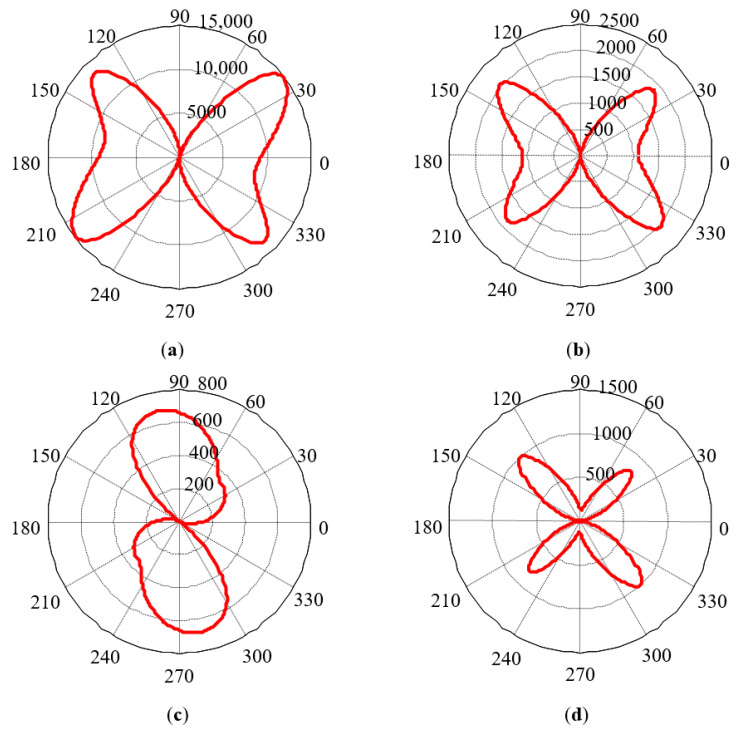
Simulation results of MCE in 4 materials: (**a**) 30SQG120, (**b**) B50A470, (**c**) X60, and (**d**) X70.

**Figure 14 sensors-21-03330-f014:**
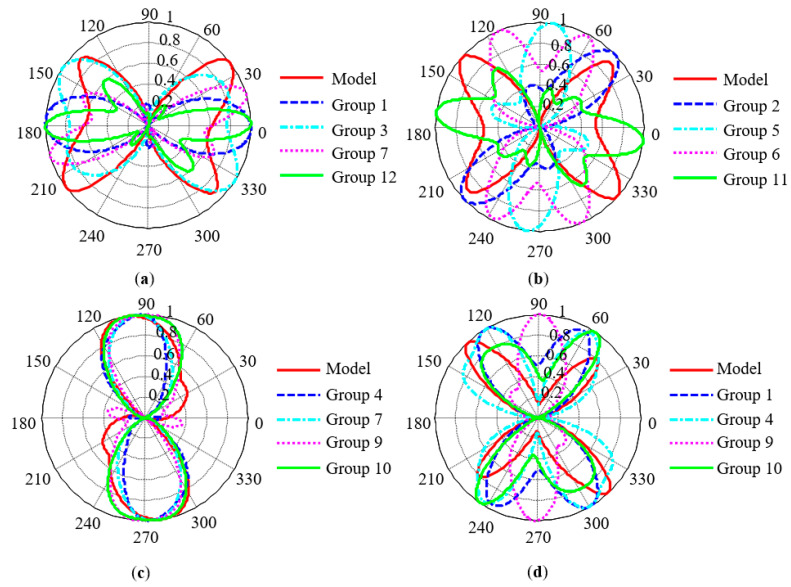
Normalized comparison between experimental results under different detection parameters and simulation results: (**a**) 30SQG120, (**b**) B50A470, (**c**) X60, and (**d**) X70.

**Figure 15 sensors-21-03330-f015:**
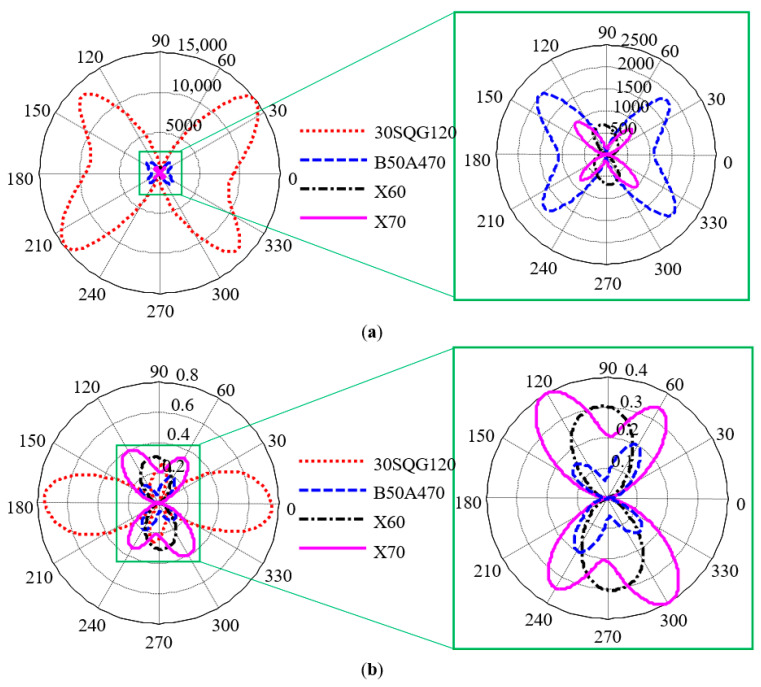
Comparison of experimental results and simulation results under fixed detection parameters: (**a**) simulation results and (**b**) experimental results.

**Table 1 sensors-21-03330-t001:** Size parameters of the U-shaped cores.

Electromagnet Numbers	Parameters (mm)			
	A	B	C	D
CX-1	120	80	60	30
CX-2	60	35	30	15
CX-3	30	35	10	10

**Table 2 sensors-21-03330-t002:** Detection parameters.

Detection Parameters	Electromagnet Numbers	Excitation Frequency/Hz	Excitation Field Amplitude/V
Group 1	CX-1	20	4
Group 2	CX-2	20	4
Group 3	CX-3	20	4
Group 4	CX-1	20	1
Group 5	CX-1	20	2
Group 6	CX-1	20	3
Group 7	CX-1	20	5
Group 8	CX-1	20	6
Group 9	CX-1	1	4
Group 10	CX-1	10	4
Group 11	CX-1	50	4
Group 12	CX-1	100	4

**Table 3 sensors-21-03330-t003:** Number of grains determined by EBSD microtexture measurement.

Material	Numbers of Measuring Points	Numbers of Grains
30SQG120	34,700	2
B50A470	216,250	1447
X60	216,750	3016
X70	216,250	1887

**Table 4 sensors-21-03330-t004:** Characteristic parameters of magnetic anisotropy obtained by two methods.

Parameters	Method	30SQG120		X60	X70	
*θ*_y_/°	Simulation	37	136	101	134	45
	Experiment	24.59	146.15	99.2	120.32	57.30
*θ*_n_/°	Simulation	82		151	0	
	Experiment	92.29		1.15	0.72	

## Data Availability

Not applicable.

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
