# Peer review of "Evaluation of the Magnetocrystalline Anisotropy of Typical Materials Using MBN Technology"

_sensors, 2021, doi:10.3390/s21103330_

Round 1

Reviewer 1 Report

Dear Authors,

The paper seem interesting, however before review full version, with figures, is needed

Author Response

First of all, I am very sorry that you did not see any figure in the file due to my negligence. The figures were not displayed completely because of the file is too large. The word version of the paper is in the attachment. If it is convenient for you, please provide valuable suggestions and comments. Finally, once again, I am deeply sorry for the delay in your work.

Reviewer 2 Report

Although I could not see any Figure in the pdf file of the manuscript, its content shows publishable information. However, some minor issues must be taken into account:

  1. All equations form (1) to (15) must use the notation, for example: E(eta, phi) or E...
  2. Eqs. (3) to (7) must be appropriately cited. 
  3. Line 304: 
    Fig. 12. Orientation imaging diagrams of 4 materials
    shoud be ..Orientation maps...
  4. How many grains were used to measure EBSD? 

Reviewer 3 Report

The authors present a model to tackle the angular dependence of the magnetocrystalline anisotropy energy (MCE) in steels The magnetic anisotropy has a crucial influence of the materials properties. The authors develop a simple model to describe the MCE in a polycrystalline material, i.e. depending on the grain orientation.

With the help of magnetic Barkhausen noise signals  which are related to the MCE they obtain the experimental input for the study.

Though the study is interesting I think some points should be discussed and clarified  before the manuscript can be published. 

On page 3: Eq. 4 … and the MCE is also directly related to the crystallographic texture…

I bit more explanation of the relation in Eq 4 is needed. Even if the MCE is related to the texture it is not completely obvious that it is directly proportional to EMBN

Page 4: Motivate the choice of the parameters K1, H, Ms

Page 4 , Fig. 3:  what does the straight line in the figure stands for? Are the stars the calculated points? What is described by delta?

Page 5: Has the grain orientation  g1 be chosen arbitrarily?

Page 14: Section 5.2 needs some reworking. Especially the discussion Fig. 15 is not clear me. Looking at the left hand side of the picture the angular dependence of 30SQG120 look very different in theory and experiment while for the silicon steel (B50A470), which should not be well explained by the model, he angular dependence is at least similar even if the magnitude is different. Some more explanations are needed. 

Though the paper presents a nice idea the authors are sometimes sloppy in presenting the results. 

In order to make the paper readable and attractive also for the none expert reader the authors should rework it including the above remarks.
